# Some Remarks on Colloid Stability: Selected Examples Taken from the Milk Chain for Food Prepares

**Camillo La Mesa [1],\* and Gianfranco Risuleo [2],†**

[1]   Department of Chemistry, Cannizzaro Building, La Sapienza University, Piazzale Aldo Moro 5, 00185 Rome, Italy

[2]   Department of Biology and Biotechnology Charles Darwin, La Sapienza University, Piazzale Aldo Moro 5, 00185 Rome, Italy; gianfranco.risuleo@uniroma1.it

\*   Correspondence: camillo.lamesa@uniroma1.it

†   Retired.

**Abstract:** Different forces play key roles in the stability of food colloid dispersions. The focus here is on those controlling attraction and/or repulsion, which concur to stabilization, phase separation, coagulation and are quite evident in water-based systems. The combination of attractive and repulsive forces favors or hinders the association of colloid entities; such processes are often met in food technology. The above processes depend on the forces at work and colloid concentration in the medium (i.e., on interparticle distance). Worked examples deal with milk manipulation procedures, ending in cheese formation. The whole milk sequence is controlled by the combination of forces leading to aggregation and phase separation of casein and other milk components. Thereafter, one gets either fresh, for prompt consumption, or aged cheeses. The combination of attractive (van der Waals, *vdW*, and depletion) with repulsive (double layer, *DL*, but also steric) forces results in the dominance of aggregation versus dispersion modes in the milk transformation chain, which depends on the distance among colloid particles, on the amplitude of the mentioned forces, and on their decay. The combined role of double layer and van der Waals (*vdW*) forces is at the basis of the *DLVO* theory on colloid stability, which is properly modified when these forces overlap with steric stabilization and, eventually, with depletion. Steric effects are dispersive, and depletion ones favor colloid nucleation in a single phase. The milk manipulation chain is a worked example of the intriguing association features controlled by the mentioned forces (and of ancillary ones, as well), and indicates which forces favor the formation of products such as parmesan or mozzarella cheese but are not alien to the preparation of many other dairy products.

**Keywords:** attractive and repulsive forces; Debye's screening length ($1/k$); Poisson–Boltzmann (*PB*) equation; food colloids; coagulation; stabilizers; lipids; polymers; electrolytes

## 1. Introduction

The cohesive forces occurring among entities in any physical body become less relevant on passing from solid to liquid and, finally, to gaseous form. Strong forces permit vibrations around the equilibrium position, moderate to strong forces permit translational motions (in liquids), and weak ones permit free and unrestricted motions, as in gases. The same also holds in colloids, be they solid, liquid, or bubbles. Colloids move as a whole kinetic entity in a fluid: diffusion in water-based media is a typical example. The colloid state, thus, deals with dispersion of solids, droplets, or bubbles in another phase. We may observe reciprocal dispersions of solids, as in opals (remaining as such for an indefinitely long time); of solids in liquids or gases; of liquids or bubbles in a liquid; and so

forth (Table 1). Given the peculiar nature of liquids, perhaps, colloid stabilization is very important in systems where the dispersant is as such.

**Table 1.** Possible colloid dispersion modes.

| State of Matter 1st Component (Dispersant) | The 2nd Component is a Solid (Dispersed) | The 2nd Component is a Liquid (Dispersed) | The 2nd Component is a Gas (Dispersed) |
|---|---|---|---|
| Solid | opals | bitumens | foams |
| Liquid | muds | Emulsions * | bubbles |
| Gas | smokes | aerosols | no |

\* Occur only if liquids are immiscible.

Stabilized colloid particles behave as planets and sense different forces. It is not surprising, therefore, that an eminent scientist (Nobel Award in 1983) put in evidence the similarities between the colloidal domain and stellar physics [1]. Despite that, most humans consider colloids as "magical" entities, for which nature looks unfamiliar such as the gravitational laws. We try to clarify below some quintessential points of colloid stability by the same methodological approach, i.e., considering the action of different forces concurring to these processes; in words, we deal with the same mental approach that is used to account for the motion of planets.

The aim of this brief review article is to induce correct scientific attitude and to avoid consolidation of absurd opinions. The ironical sentence below is an example of the behavior to be avoided. "Reporting on the motions of earth and planets, Galileo treated the catholic church with the same wickedness that a classmate used against my beliefs. He told me '*Santa Claus does not exist, reindeer-driven sleighs neither.*' How much different I was from him! Should granny have told me that donkeys fly, I should have never replied they do not, but, presumably, I would have added that horses, mules, ponies, the seven dwarves, the twelve apostles, and our archbishop (despite his modernistic attitudes) do that. And to render granny happier, I should have told her that, in Paradise, she will have met Pinocchio and Geppetto, the Booted Cat (more properly, Le Chat Botté), the Red Feathered Young Teacher, the harmonious Smith and Queen Margret, angels, saints, and other canonically expected guests." [2]. Here, we try to persuade readers not to behave as the mad character depicted above. Figure 1 shows some items to be discussed in the following figures presented.

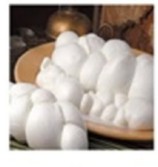 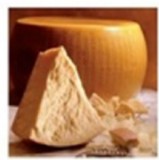

**Figure 1.** (**A**) "Braided" fresh mozzarella from buffalo cheese, typically produced mainly in central-southern Italy but possibly appreciated all over the world. (**B**) Parmesan cheese shape and wedge: the most select parmesan cheese may have undergone up to 36 months of aging in controlled atmosphere. The standard weight of a matured parmesan shape is about 40 kg, for which about 550 lt of cow milk is required.

The focus here is on food colloids, such as milk, mayonnaise, chocolate, and juices. These are sometimes obtained with the help of mechanical energy, crumbling the raw products, giving to these dispersions the desired shapes and forms, and imparting energy, later dissipated in form of heat [3]. The optimization of food colloids is of fundamental interest due to the growing demand for a wide availability of staple food, which are characterized by high quality and nutritious capacity. Raw matter quality, attention to manipulation, and long-term stabilization are relevant starting points too [4].

Below, we focus on the steps to be followed in the food manipulation chain and on the forces controlling each of them. That is why food technicians always try to optimize the following manipulation stages:

(a)   collecting raw matter;
(b)   operating with it or mixtures of the like in such a way to get the desired products of a given food chain in substantial amounts; and
(c)   obtaining the final products through maturing and storage, if these steps are required.

Cooking, drying, salting, smoking, and storage procedures have belonged to human knowledge since many thousand years [5–7]. Preparations such as anchovies in paste [8], stock cubes, and cheese [9,10] are pertinent examples; many of these products use stabilizers during the manipulation stages, and all are colloidal in nature, or are as such, at least, during some manipulation steps.

We deal with fluid, semi-fluid matrices, creams, and pastes, in other words, "*soft matter*" foods. To make such preparations reliable and nutritionally safe with "permanent" macroscopic appearance, homogeneity upon aging and retainment of adequate properties throughout different batches and seasons must be granted. Food preparations are also optimized from a chemical and biological viewpoint to fulfill due health requirements. According to the rules, and good sense too, any colloid-based food must be stable and must retain its peculiarities and taste for long times. There is also urgent need for storage that does not require conditions hardly at hand in some disadvantaged situations, when, for instance, low-temperature chains cannot be guaranteed. Many old-fashioned preparations, which are costly niche products, may give high-quality matter but might operate in small to medium mass scales, with drawbacks due to their costs, retailing, and durability.

Food manipulation gives creams, fluids, pastes, sauces, etc., which are stabilized by adding salts, lipids, some proteins, or polysaccharides [11–14]. Stabilizers are preferentially taste-neutral, biocompatible, and not too expensive. They should be from natural sources in large amounts and, compatibly, from the same materials as the products to be stabilized. Food colloids have rheological properties allowing them to flow or spread even under moderate shear conditions. In addition, their nutritional quality must fulfill all the standards required from national/international procedural guidelines [15].

We indicate below which forces are mainly responsible for food colloids stability and apply pertinent information to the case of cheese. This largely consumed commodity (there are several hundred cheese types, perhaps) undergoes many transformations from raw milk. The latter, a water-based bio-colloid, contains not only fats but also proteins (about 82% is made of different casein types, and the remaining are serum or whey proteins) [16], sugars (mostly lactose), salts, etc. Fat content depends on mammal type (Table 2). Similar considerations hold for protein, sugar, and salt content. Goat milk, for instance, has low amounts of calcium and phosphate and is enriched with their salts [17].

**Table 2.** Average fat matter percent in fresh milk of some mammal ruminants.

| Mammal Ruminant | Country | Fat % in Fresh Milk |
|---|---|---|
| Jersey cow | United Kingdom | 5.2% |
| Zebu | India, Madagascar, Sri Lanka | 4.7% |
| Brown Swiss cow | Switzerland | 4.0% |
| Holstein Friesian cow | Netherlands, FRG, Denmark | 3.6% |
| Neapolitan Buffalo | Italy | 8.3% |
| Spanish sheep | Spain, Italy, Greece | 5.6% |
| Merinos-modified sheep | Spain, Australia | 5.3% |
| Appenzell goat | Switzerland | 4.6% |
| Manchega goat | Spain, France | 4.8% |

N.B. Data reported in the table refer to average values inferred from different sources.

## 2. Surface Properties

### 2.1. Forces in Action and Species Responsible for Stabilization

Different chemical, biochemical, and physicochemical properties characterize the most common features of products and stabilizers used in food industry. Surface adsorption, among many other forces, is strictly necessary, since stabilization is almost always concomitant to interface saturation [18]. It is immaterial if such processes arise by covalent or non-covalent polymer binding. Adsorption also implies wrapping [19], steric [20], osmotic [21], electrostatic effects [22], and combinations thereof [23]. It is hardly ascertained *a priori* if stabilization results from the combination of more contributions; think of the role of *k*-carragenans [24] or lipids [25] as stabilizers. The former are linear sulfated polysaccharides obtained in seaweeds (by *Chondrus crispus algae*, also known with the vulgar name of Irish moss). They have been in use since the XVI century but were known in China as early as the 6th century BCE. Nowadays, their main application is in dairy and meat products for their ability to establish strong bonds with food proteins [26]. Carragenan is a vegetarian and vegan alternative to gelatin and is also used to replace the latter in confectionery and ice creams [27,28].

We mention below the forces responsible for the making and stability of cheese from the milk food chain. Among them, surface adsorption, van der Waals (*vdW*), and steric and electrostatic forces are worth considering; others may be also present. We start with surface adsorption, which is somehow connected to steric stabilization; in fact, adsorption is a prerequisite for steric effects to occur. Later, some generalities on electrostatic and van der Waals forces will be given. The rationale inherent to the above sequence is evident. We stress again that steric effects arise if polymer surface adsorption has occurred and are ancillary to surface energy, exactly as Sancho Panza reluctantly follows the noble knight Don Quixote.

### 2.2. Surface Properties

Adsorption onto colloids occurs through surface binding of stabilizers. These resulting forces are similar to those allowing surfactants to adsorb at fluid surfaces. Surface tension is due to the gain associated to chemical transfer from the bulk to an interface. It is made explicit by the Gibbs isotherm, linking the energy of adsorption to the surface tension, $\gamma$, and differential area, $dA$, as follows:

$$DG_{ads} = \gamma dA \tag{1}$$

The above relation is at the basis of surface activity, mostly to the adsorption of surfactants and lipids at air–liquid interfaces. The behavior of surfactants and lipids is well known: interested readers in this particular point may refer to classical compilations [29–31]. As a matter of fact, however, the equation applies to all substances adsorbing at interfaces. Note that outdated efforts to determine the molecular weight of lysozyme used surface tension (with appreciably good results, indeed) [32]. Let us consider in what follows the surface behavior of bio-polymers and assume that, if the surface tension of their aqueous mixtures is lower than that of the solvent, polymers adsorb. The surface tension, $\gamma$, of binary systems (and not only, perhaps) depends on composition, nature of adsorbed species (Figure 2), and temperature, $T$. The latter fact makes it possible to change surface composition from a polymer-rich to a polymer-depleted state, provided such a $T$-dependent behavior is fulfilled somewhere (Figure 3). Stabilizers adsorb onto colloids thanks to surface tension. Such an energy contribution is due to the transfer of chemicals from the bulk to interfaces and is expressed by Gibbs adsorption isotherm. If the surface tension of a polymer (supposedly in melt form) is lower than that of the solvent, the first is adsorbed on the particle surface. For flat and fluid interfaces, the Gibbs equation is defined as follows:

$$d\gamma = -\Gamma_2 d\mu_2 = -\Gamma_2 RT d\ln a_2 \tag{2}$$

where $d\gamma$ is the differential surface tension with respect to the solute and $\Gamma_{2\,is}$ its excess concentration per unit area. Equation (2) indicates that $\gamma$ is inversely proportional to the "partial molal area" $(1/\Gamma_2)$

of the solute. The latter depends on the solute activity, $a_2$, and vanishes if there is no room for adsorption, that is, when the surface is saturated by the solute. Originally intended for flat fluid surfaces, Equation (2) also applies to drops and emulsions. Note that Equation (2) must be modified in the case of multicomponent systems, where minima in the surface tension vs. $ln a_2$ plots may occur. Such features were observed in aqueous surfactant solutions doped with very tiny amounts of long-chain alkanols.

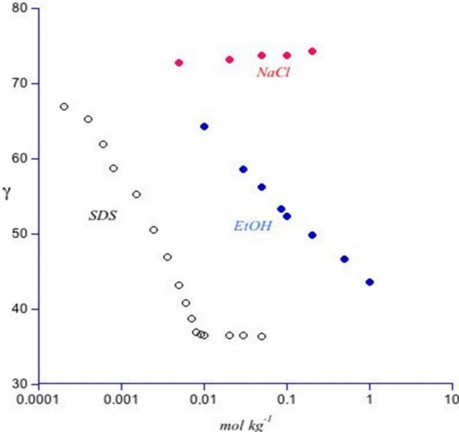

**Figure 2.** Variation of the surface tension of aqueous mixtures as a function of composition, in arbitrary scale: data refer to NaCl, full line; ethanol, empty circles; and a generic surfactant. When NaCl allows for increasing the surface tension, the other two species decrease it. Surfactants saturate the surface phase of water above a critical threshold, termed *cmc*.

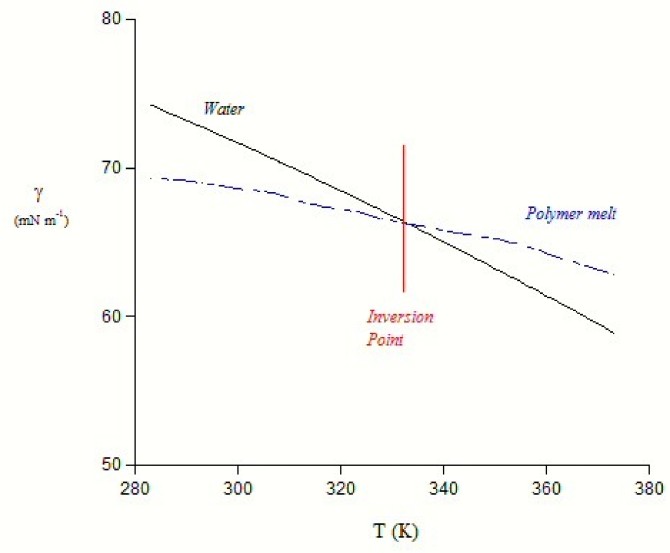

**Figure 3.** Surface tension of a water/polymer mixture as a function of temperature, in K: the vertical line in the center indicates the inversion temperature, $T_{inv}$, below which the polymer is surface-depleted. At T below the "inversion point", the polymer prefers being in the surface phase. If a solid is present, surface stabilization occurs from this point onward.

Complications arise when stabilizers adsorb onto porous solids [33]. In the above eventuality, the configuration of a flexible polymer interacting with a surface somehow depends on *T*, so that the given polymer is adsorbed or repelled. Theoretical studies on this line consider polymer adsorption on either smooth and porous solid surfaces [34,35]. In some sense, steric stabilization behaves in analogy to the action of a surface-anchored spring exerting a repulsive force per unit area. Some remarks on this point are outlined in the forthcoming section.

### 3. Steric Stabilization

Such a contribution is hardly understood on primitive grounds. It is known that the structure and properties of long polymer chains, grafted or colloid-bound, control steric effects. Steric stability is considered entropy-controlled; indeed, it can be enthalpy- or entropy-driven [36]. It contains many other terms, such as osmotic, excluded volume, conformational, electrostatic, and combinations thereof. Due to the lack of a univocal theory, current models in the field use molecular dynamics or other types of simulation procedures.

To account for steric stability, we consider flat and parallel surfaces at indefinite distances apart; in between, polymer and solvent are present in due amounts. Adsorption and spreading take place on both surfaces until they are saturated by the polymer. Expansion, compression, mixing with the solvent, and interpenetration stages take place on both surfaces until full equilibrium conditions are attained. One is considering forces acting among flat surfaces (Figure 4). Such a behavior resembles what we observe in surface forces apparatuses [37].

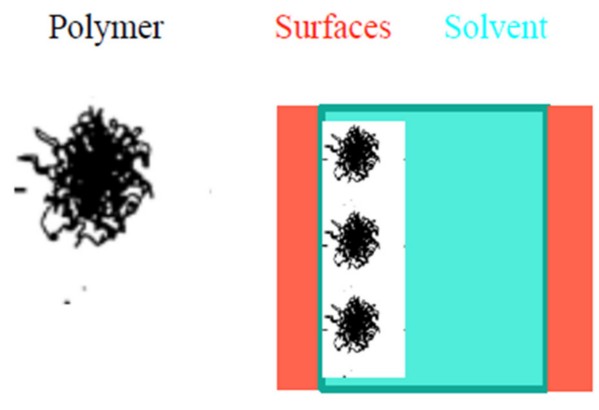

**Figure 4.** Covered surfaces, in red at a distance X apart with the polymer and solvent in between: polymers mix with the solvent, light blue; thus, surfaces saturate. Compression of solvated polymer films occurs, and interpenetration takes place. This is why polymer chains are not drawn in direct contact with the surface, as they should be. Each step has its own energy contribution, and the final stabilization is the sum of these terms. The distance among surfaces is, by definition, much higher than the polymer size.

Steric stabilization is evaluated by mean-field theories based on the interactions between two plates covered with densely grafted polymer chains; the concept "*dense*" is, perhaps, elusive. The interaction potential depends on the polymerization degree and increases with it, with the stiffness of stabilizing chains, the solvent quality, and number density of adsorbed chains. Steric stabilization is hereafter indicated as $\Xi$, energy, $E_{att}$, on the amount of grafted polymer, $\sigma_{Polym}$ ($C_{polym}/A_{part}$), and on deviations from $\theta$-temperature conditions, $d\theta$. More refined approaches are at hand, but we use here a better primitive model, written as follows:

$$\Xi = f(E_{att}, \sigma_{Polym}, d\theta) \tag{3}$$

The minimum of $\Xi$ in the $E$, $\sigma$, and $\theta$ space implies thermodynamic stability. For weak attractions, curves calculated by Equation (3) display a secondary minimum. As a rule, if $E_{att}$ is lower than other terms, thermodynamic stability conditions are met. For strong inter-particle attraction, in the presence of good solvents for the polymer chains, the potential displays primary and secondary minima,

separated from each other by a maximum. The stability in the latter case is kinetic in character [38]. Other effects are concomitant to steric stabilization, such as osmotic, depletion (due to concentration gradients sensed by polymer chains located between two particles), and electrostatic (for charged polymers). Sometimes, it is necessary to reduce steric stability and to favor nucleation. A pertinent case is due to rennet, cutting the proteins located outward from casein clusters and facilitating the cheese seeds onset.

## 4. Electrostatic Contributions

Electrostatics influence food stabilization, and knowledge is, thus, required of some aspects of colloid stability in terms of *DLVO* theory [39,40], combining *vdW* and *DL* forces. The resulting energy is due to the overlap of such forces; the former is always attractive, while the latter is repulsive. Refinements and implementations of the theory are available [41–44]. *DLVO* theory explains why colloids disperse, attract, or coagulate depending on the experimental conditions.

A combination of the mentioned forces controls interactions; stability can revert to instability upon very tiny changes in the control variables, such as added salt, pH, or screening the surface charge density in some other way. Similarly, charged surfaces undergo long-range repulsions and the barriers keeping them apart can be some $K_BT$ units high. If the electrolyte content increases, a secondary minimum occurs. In such conditions, $\sigma$ (the surface charge density) approaches zero, the repulsive forces minimize, and attractive terms dominate (Figure 5). Colloids are characterized by a given size and electrical surface potential. Uncharged colloids coagulate, whereas surface charges density avoid it. Stabilizers impart a "permanent" surface charge density. Let us consider two surfaces as specular images, each of the other, both with high $|\sigma|$ value. In consequence of that, surfaces repel. Repulsion depends on the potential, $|\Psi|$, which exerts a long-distance effect scaling with $kD$. $D$ is the distance, and $1/k$ is Debye's screening length. The effect has the same meaning as that between two planes bearing equal $\Psi$ values.

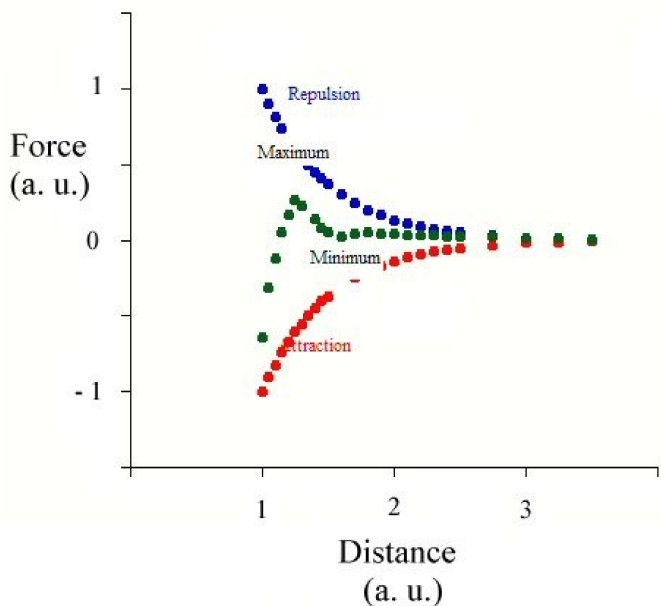

**Figure 5.** Schematic view of the forces acting among two colloids or surfaces vs. distance: repulsive (light blue), electrostatic, and *vdW* (red) forces are considered. The result (in green) is the sum of these contributions. The minimum in the curve refers to stability. The position and amplitude of either maxima and minima in the green curve depend on the ionic strength of the medium.

Potentials decay is as follows:

$$\Psi(x) = \Psi° \, exp^{\,-kD} \tag{4}$$

where $D$ is the distance from a virtual charged surface of potential $\Psi^\circ$ (Figure 6). The equation refers to the interaction between two surfaces having the same electrical potential. It is written as:

$$\nabla^2\Psi = d^2\Psi/dx^2 + d^2\Psi/dy^2 + d^2\Psi/dz^2 = -(\varrho/\varepsilon\varepsilon^\circ) \tag{5}$$

where $\varrho$ is the ion number density, and $\varepsilon$ and $\varepsilon^\circ$ are the permittivity of vacuum and dispersant, respectively. The electric field in Equation (5) is radial; so, we consider its components only along axis $x$ and rewrite it as follows:

$$\nabla^2\Psi = d^2\Psi/dx^2 = -(\varrho/\varepsilon\varepsilon^\circ) \tag{6}$$

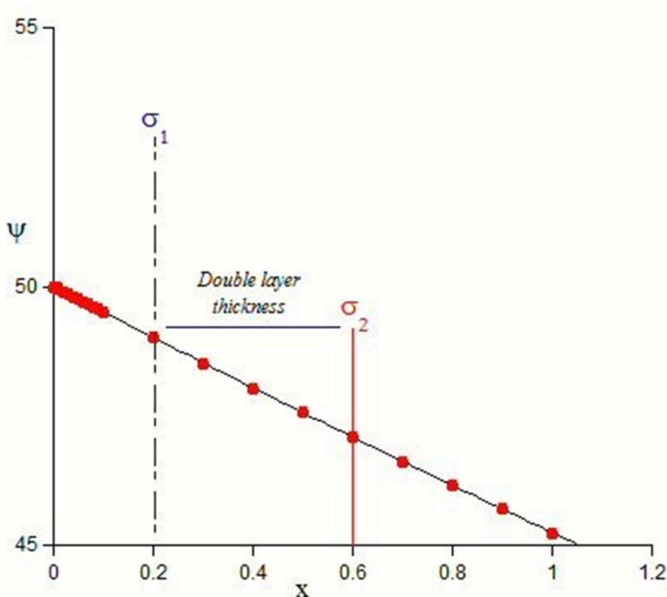

**Figure 6.** The decay of electrostatic potential, $\Psi$, and the surface charge density, $\sigma$, versus distance, $x$: the distance among the points having charge density equal to $\sigma_1$ and $\sigma_2$, the double layer thickness, or Debye's screening length, depends on the medium ionic strength.

As for the statistical energy term, the Boltzmann's law for distribution of charged species in a medium is written according to the following:

$$c_i = c_i^\circ \, exp^{\,-(ze\Psi/KBT)} \tag{7}$$

where $c_i$ is the local concentration of the $i$th ion, $c_i^\circ$ is its actual one, $ze\Psi$ is the energy associated with the electric field for an ion of valence $z$, and $K_B T$ is the thermal energy term. Equation (6) is the ratio of electrical to thermal energy and determines ion distribution around charged entities depending on electric fields and thermal motions. Each ion contributes to the electric field:

$$\varrho = \varepsilon(c^+ - c^-) = \varepsilon c_i^\circ (exp^{\,-(ze\Psi/KBT)} - exp^{\,(ze\Psi/KBT)}) \tag{8}$$

In that form, the equation represents the charge density due to ions in excess, be they >0 or <0. If $|(ze\Psi/K_B T)|$, the difference among exponentials transforms in hyperbolic form ($exp^x - exp^{-x} = 2sinhx$), and is linearized. In that case, Equation (7) indicates a "linear" perturbation regime, determining the electric contributions to the energy. Advantages inherent to the above linearization procedures are substantial.

The charge density, $\varrho$, is related to surface potential, $\sigma$, which depends on $var\Psi$. Links between $\varrho$, $\sigma$, and $\Psi$ are expressed as follows:

$$\sigma = -\int \varrho dx \tag{9}$$

$$\sigma = (2n°\varepsilon K_B T/\Pi)^{1/2} \, sinh(ze\Psi/K_B T) \tag{10}$$

where the meaning of the symbols is as before. From Equation (9), we relate the system energy to electrical potentials, according to the following:

$$\Delta G = -\int \sigma d\Psi \tag{11}$$

Proper combination of all the above forces allows for analyzation of experimental data and forecast of the expected energy contribution.

## 5. More Fundamentals

### 5.1. The Fundamentals of Food Colloid Stabilization

To show how much the overlap of these effects is important in the food chain, we consider milk transformation. Although that process is well acquainted, the underlying theory is not fully known. We do not consider transient stirring; in the above case, dispersions revert to the original conditions in the time lapse required to dissipate the mechanical energy imparted and released in the form of heat. Here, the focus is on "stable" food dispersions, having shelf-lives of weeks or months, that is, on systems in thermodynamic, kinetic (we do not worry so much about such differences at the moment), or equilibrium conditions!

Animal-based foods and our bodies contain from 55% to 75% water, depending on the tissue. What remains is made of fats, lipids, sugars, and proteins, that are associated in liquids, gels, liquid crystalline phases, pastes, and amorphous or semi-solid matrices. Food tissues bear a consistency which depends on the organization of colloid entities in a tissue. From this evidence, the generalization comes out that all animals, humans, and vegetables consist of different colloid entities, which eventually coagulate.

Coagulation does not occur among charged particles because of repulsive electrostatic forces. This holds true in oil droplets stabilized by charged lipid layers. Low electrolyte amounts ensure droplets repel each other. If $\varrho$, $\sigma$, or $\Psi$ are small, the energy barrier among particles (always related to $ze\Psi$) is low and a marked tendency to coagulation is observed. The limit at which these phenomena occur is the so-called flocculation threshold. An increase of salt decreases $\Psi$ and ensures coagulation. Thus, as $\sigma$ tends to 0, $DL$ forces are null, electrostatic terms vanish, and particles attract. When the $pH$ or the ionic strength, $I$, change, food colloids aggregate or redisperse (depending on the experimental conditions). Steric stabilization forces, which act as a dispersive effect, overlap with electrostatic forces and impart to the dispersions a substantial stability (Figure 7).

### 5.2. The Case of Food Colloids

Peculiar are milk manipulation methods ending in cheese formation. The whole process is controlled by the relative amounts of fatty acids and glycerides, existing in the form of oil droplets, plus micelle-forming casein, proteolytic enzymes, salts, lactose (the main milk sugar component, later transformed in lactic acid), and so forth. The process ends when aggregation/gelation occurs, is governed by $T$, enzymic activity, changes in $pH$, or presents salts, or combinations thereof. Many are the routes leading to coagulation, and forces may favor a seed-like or fibrous organization. This is why we may get *parmesan* (with its hard to the teeth and exhibits a typical "*sandy*" consistency) [45], *provolone* and *mozzarella* cheese (both consisting in fibrous structures) [46], or *ricotta* (soft and flaky), which may largely differ from the other in terms of taste, smell, consistency, and stiffness.

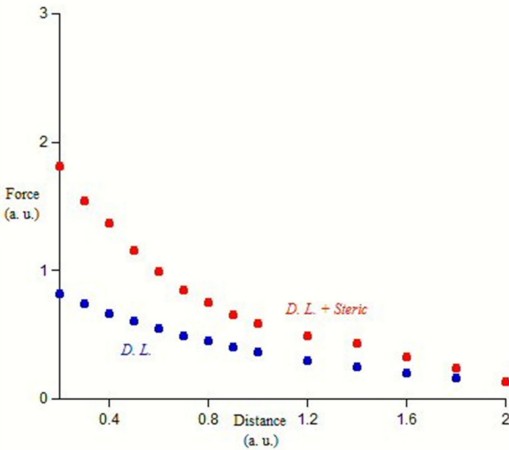

**Figure 7.** Overlap of purely electrostatic terms, in light blue, with steric stabilization, in red, forces at short distances: forces are in arbitrary units. It is evident, therefore, that the elimination of steric terms largely reduces the system stability.

Differences among the above products find an origin in the fact that stretching, eventually combined with heating, deforms cheese seeds and imparts a preferred direction along which fibers are oriented by the action of cheese weight and are preferentially oriented in the stress direction. Both stretching and subsequent deformations are somehow due to the plasticity of raw cheese paste.

Although casein micelles are charged, added salt does not necessarily ensure coagulation or formation of "cheese seeds". In fact, such micelle-like entities are stabilized by steric forces, not allowing them to come in deep contact and to coagulate. This fact is put in evidence in Figure 7, when steric and *DL* forces overlap. Steric stabilization, thus, counteracts attractive forces. The effect is destroyed by *rennet*, a proteolytic enzyme cutting the *k*-casein parts facing outward micelles. In the early cheese-making steps, *pH* values close to 4.8 activate/deactivate the hydrolytic enzymes, for which activity also depends on *T* and salt, eventually [47]. Sometimes, the above enzyme is replaced by *twistle* or *fig* tree and branches, the extracts of which have very pronounced proteolytic capacity. *Rennet* contains different enzymes, all produced in the stomachs of young ruminants; its key component, *chimosin*, is a protease curdling milk casein; and *pepsin*, *phosphatases*, and some *lipases* are present too.

*Chimosin*'s main action consists in the cleaving of *k*-casein chains [48]. Cleavage causes casein residues to stick to cleaved molecules of the like and to form a network. *Rennet* action, thus, (a) eliminates stabilizing moieties among casein micelles and (b) makes it possible for such entities to form a network clustering fat droplets and minerals. The latter entities are sometimes termed "*cheese seeds*". The structure of these network depends on several variables. Size and elasticity of seeds depend on network size; the same holds for their consistency, aging modes, and spreading facility (if any).

*Rennet* better clusters casein in the presence of calcium and phosphate, occasionally added in the cheese making of goat milk, which is calcium phosphate-poor [49]. The casein protein network traps other components of milk, mostly fat droplets. *Rennet* also separates milk in solid curds (in cheesemaking) and liquid whey, and so do its substitutes. Also relevant is salt concentration and valence; calcium, for instance, favors aggregation compared to sodium or potassium. Thus, the presence of ions and screening of electrostatic forces are relevant, once proteolytic activity has gone to completion. To clarify such aspects, we introduce below a simple electrostatic approach to stabilization and show how much the latter is relevant in milk-based formulations.

As mentioned above, repulsive forces are electrical, steric, or osmotic. We consider first the role of the former in favoring/disfavoring phase separation. Steric, osmotic, and *DL* contributions counteract *vdW* forces and shift the coagulation threshold towards high concentrations. These features, observed in some stages of cheese making, are summarized below. In *DLVO* theory, *vdW* forces are always combined with *DL* ones. For two bodies at constant *T*, the interaction energy, $E_{int}$, depends on the reciprocal distance, *D*. At high *D*s, $E_{int}$ and *DL* terms approach zero; in such a regime, the sum of

all contributions vanishes. No energy gain is associated to the interactions between particles at long distances. The presence of a primary minimum, occurring at short distances, and a secondary one are met when distances are lower. The secondary minimum shifts in proportion to *I*; it is separated from the primary by an energy barrier, the height of which depends on activation energy to coagulation. The secondary minimum is some $K_B T$ units high and shifts to lower distances in proportion to *I*. The figure indicates that the tendency to coagulate is represented by the overlap of energy curves on *vdW* ones [50].

The maximum therein depends on the activation energy to coagulation, $E_{att}$. The role of ionic strength, $I = 1/2\ \Sigma_{i=1}\ c_i z_i^2$, is put in evidence by considering what happens when the potential among surfaces having a fixed number of charges per unit area is screened by due amounts of added salt. In distilled water, $\Psi°$ rapidly increases with ion concentration. Neutral electrolytes (some mmol kg$^{-1}$) have a buffer effect on $\Psi°$. Since most food preparations contain substantial amounts of salt, it is clear why the region where the potentials are effective is between 25 and 80–100 mV in modulus. For $\Psi < |25|$ mV, samples tend to coagulate; if $\Psi > 100$ mV, most counter-ions adsorb on the particle surface and minimize repulsions. Thus, changes in $\Psi$ ensure dispersion, aggregation, or sedimentation [51].

In the calculations, we may combine all the mentioned forces in the generalized relation:

$$E_{tot} = \Sigma_{i=1}\ E_i\ exp^{-kiD} \tag{12}$$

where $E_i$ is a given energy mode. More complex formulations for the system energy scale such as $1/D^n$ (with $n \geq 3$) have been widely described [52]. These energy terms conform to short-distance decay modes. Thus, repulsion rapidly decreases with distance. Attraction, conversely, is governed by *vdW* terms responsible for phase separation. On this regard, the difference between yogurt and cheese coagulation clarifies what physical forces govern the onset of these materials, taking origin from the same natural source.

Electrical potentials in colloids are measured by $\zeta$-potential, a distance *d* apart from the outer limit of the slipping plane; by electrophoretic mobility; or by Laser–Doppler facility [53,54]. The decay of such $\zeta$-potentials with *pH* and/or *I* can be determined, and the surface charges are titrated, with subsequent coagulation or re-dispersion (Figure 8). The salient point in the figure is the "nominal" point of zero charge, *p.z.c.* For proteins, the *p.z.c.* corresponds to the isoelectric point [55–57].

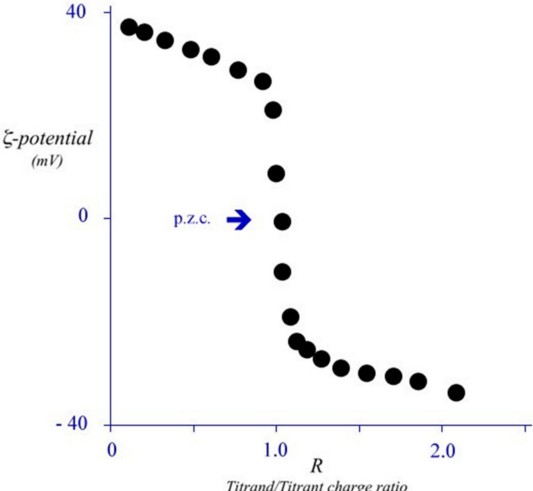

**Figure 8.** $\zeta$-potential, in mV, vs. versus the titrant to titrand, *R*, ratio: if $R \cong 1$, the $\zeta$-potential is 0. The curve is grossly symmetrical with respect to the *p.z.c.* the inflection point in the graph. Data refer to sunflower oil dispersed by *DPDC* (Dipalmitoyl Phosphatidyl choline), in 0.0250 aqueous NaCl, at 35.0 °C. The titrand/titrant ratio, *R*, depends on pH.

## 6. The Cheese Making Sequence

Actual cheese-making sequences are not different from what is historically known. They consist of 5, sometimes 6, stages:

1.  Milk coagulation by *rennet* (sometimes assisted by tiny amounts of citric acid) added to milk kept at mild temperature, 36–39 °C. This step implies optimization of the enzymic activity of *rennet* and, in consequence of that, favors eliminating steric stabilization by cutting the polypeptide chains protruding outward from casein micelles.

2.  Curd breaking and serum discharge jointly imply a phase separation of colloids. A heating stage at 50 °C is required to completely discharge and favor the onset of cheese lumps.

3.  Curd is left maturing in acid serum for some hours. In this stage, solubilization of calcium salts and de-mineralization of cheese paste occur. The paste becomes plastic (is a reverse phase separation), giving rise to "*pasta filata*" (also termed *cooked pasta*), and is reduced in fibers by the action of gravitation or by simple stretching, as happens for mozzarella and provolone. In the absence of shear forces, which favor the formation of fibers, cheese grains collapse.

4.  Salting is made in dry or wet conditions, depending on the needs. Ion valence and content are relevant in this step. Salting implies water loss due to the onset of osmotic gradients outside the cheese peel, which contracts, deforms, and progressively becomes harder.

5.  Cheese maturing is usually performed at 3–8 °C and 85–90% humidity. It helps to give cheese due consistency and appearance. It implies further water loss. Maturing depends on the cheese kind obtained, and the same holds for working conditions. In niche products, mildewing, coverage with straw, or chestnut leaves (which are rich in tannins) are sometimes used. Sometimes, mildewed conditions are induced on the cheese peel or in its interior. In the latter eventuality, a wide variety of "*blue cheese*" types may occur. These can be stiff (*Stilton*) or creamy (*Gorgonzola*).

## 7. Conclusions

The transformation of raw foods that can be properly stored is well known since the very early stages of human civilization. It is urgent, nowadays, to optimize these transformations in such a way that the products obtained by any food manipulation chain are safe and can be stored for long times. That is the reason why efforts are currently done in the search to optimize the properties of primary matter and subsequent formulations in real food. To do this, advanced scientific approaches must supplement what is done in current use procedures. These efforts actually find considerable attention, due to the fact that the amount of available food resources is not much higher if compared to the number of humans worldwide. Transformation of primary matter, thus, requires optimization in such a way to get food(s) with good nutritional quality, that are safe, and that have substantial attitudes to be stored even in non-optimal conditions. To proceed along this line, we rely on incontrovertible facts. Most primary raw matter is usually transformed in products having colloid nature. The latter takes the form of pastes, creams, and semi-solid matter, for which the consistency spans from that of yogurt to that of hard cheese.

Theoretical approaches based on the fundamentals of colloid chemistry, thus, find a rationale. By them, we can evaluate the role of the forces mentioned above in determining optimal sequences and the hierarchy of the procedures giving rise to a given colloid organization mode. In doing this, we operate exactly in the same way that was the former used in optimizing a given product compared to others. The only substantial difference is that we may control each step.

We are aware of the fact that the contributions to the overall system stability due to the forces we have indicated before are substantially different, in modulus, from each other. In addition, some of them are attractive (as van der Waals) and others are repulsive (as double layer ones). Evaluation of all these effects can, thus, be cumbersome. Furthermore, the effects due to each of them depend on distance, that is, on the concentration of the dispersed matter and scales according to different power law modes. Think, for instance, to the substantial difference among the aforementioned *vdW* and *DL*

forces! It is also important to check if it is possible to pass, and in which way, from finely dispersed to a coagulated state, as experimentally observed in some stages of food manipulation. All these aspects are relevant in forecasting experimental results, if they are well modulated.

The calculus procedures and methodological approaches suggested here are made possible only by considering the mentioned terms in a sort of hierarchical scale, that is, progressively adding one more term of those of minor relevance. This procedure will help define the real contributions pertinent to each of the energy modes that are present in any food preparation chain.

We regret if our technical knowledge of the cheese production chain is limited to the role of appraisers and knowers. That is why we essentially focused on the forces in action for each milk manipulation step, without entering into much technical details on the complex art termed cheese-making. Being not capable to enter into technical details, we conform to the disconsolate opinion that a French Politician of the last century, Charles De Gaulle, used upon retirement from the active policy: "It is impossible to govern a country worldwide famous for producing well over 400 different cheese kinds".

**Author Contributions:** The authors jointly contributed to the manuscript. G.R. took care of the biological-biochemical parts; C.L.M. dealt with the physicochemical bases. Writing, review, and editing were made by both authors. All authors have read and agreed to the published version of the manuscript.

**Funding:** This research received no external funding.

**Acknowledgments:** C.L.M. wishes to acknowledge a dear friend, Giuseppe Ranieri, formerly at Calabria University, for providing a copy of an old dated article by Chandrasekkhar [1]. Chandrasekkhar's mathematical treatment of stellar evolution gave rise to many theoretical models on the early evolutionary stages of massive stars and black holes. And on colloids too, mostly on theories of diffusion in porous media. We also thank Lorenza Suber, formerly at CNR, for suggesting Arbasino's book [2]. As to Ref. [2], the late Alberto Arbasino passed over in March 2020 at the age of 90. He was a brilliant and argumentative writer, and a sarcastic polemicist. We translated this piece, regretting not to have been fully capable to transmit its temper. Pinocchio has possibly a world-wide resonance, and the same holds for Le Chat Botteè. The Red Feathered Young Teacher may be less known. However, she is a main character in a short story of the book "Cuore" a popular reading for primary school children (but not only them) by the Italian XIX century novelist, painter, and poet Edmondo De Amicis.

**Conflicts of Interest:** The authors declare no conflict of interest.

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
