# Peer review of "Some Remarks on Colloid Stability: Selected Examples Taken from the Milk Chain for Food Prepares"

_colloids, doi:10.3390/colloids4040058_

Round 1
Reviewer 1 Report
The aim of this manuscript is to make a link between two scientific communities (Food Science and Physicochemical approach) and is very usefull to help mutual understanding of soft matter problems around Food prepares.
The changes in the manuscript are in accordance with my precedent request and I think that this manuscrit can now be published.
I just want to mention to minor remarks:
* in &5.2: in the following sentence it seems that a subject is missing after the ";".
The secondary minimum shifts in proportion to I; is separated from the primary by an energy barrier, whose height depends on activation energy to coagulation.
* in the same &, the number of the figure must be indicated:
The figure indicates that the tendency to coagulate is represented by the overlapping of en- ergy curves on vdW ones [50].
* When I open the pdf file, square appears instead of the symbols (eq 10, eq 11 end in the text). This point should be carefully checked.
Reviewer 2 Report
The work by La Mesa and Risuleo presents a very nice review of colloidal stability focusing its interest in dietary milk products. The work is well-written and the discussion is sound. I have only a very minor commnent:
-Authors discuss the impact of the adsorption in surface tension. However, they discuss the process considering the adsorption of a single component when most of the products present complex mixtures of polymer and surfactants. Can the author introduce some discussion on such aspect?
Author Response
Please see the attachment.

This manuscript is a resubmission of an earlier submission. The following is a list of the peer review reports and author responses from that submission.
Round 1
Reviewer 1 Report
The approach is interesting: technofunctional application of milk (cheese making) is presented together with physical bases of the acting forces between colloidal “particles”. Both aspects are clearly described, giving bases to the reader to learn more. I particularly like the style of the beginning of the article: it facilitates access to physical concepts, sometimes off-putting or intimidating for non-physicists. On another side, the description of cheese elaboration is precisely described.
Hence, this paper is interesting either for soft matter scientists discovering food science, either by food scientists discovering soft matter basis. For that reason, it is important to keep the link between the two parts and some improvements could be made on this point. This is, indeed, my major remark. Other remarks are minor one.
Major remark: improvement of the link between the two parts
1) to help the non-physicist reader to keep an interest in the "physical" part of the article and to give a concrete example to the reader non familiar with food science, a short example of the effect of each force on a dairy product could systematically be given at the end of each paragraph (as it is done in §1.3 with caseins and the steric effect). So, §1.2 could be ended by a brief recall of protein adsorption on hydrophilic/hydrophobic surfaces in foams, §1.4 could briefly discuss the impact of salt on foams, emulsions etc…
2) Symetrically to the precedent remark, the descriptions of food process should systematically, not only refer to the forces at play but specify on which physical parameter, in which formula, such parameter acts (i.e. impact of salt on Debye length, therefore on the potential, formula (3) etc…) I think that it would greatly help the reader to link the two parts.
For example: Line 247-248: “an increase of salt decreases Psi and ensures coagulation” is too compact. The authors should include some explanation to precise that the effect of salt is to reduce electrostatic forces (refer to the right formula) etc... The idea would be to make clearly the link between the part of the paper describing the physical forces and the effect on these forces by physicochemical parameters.
Idem for lines 339-355: it should be very interesting to make the link more explicitely with physical forces at each step of cheese-making, with eventualy refering to physical terms in the formula (when it is possible). It would help the reader non physicist to understand on which physical term a given parameter (ionic force, acidification etc..) acts.
Line 277-278 “steric stabilization…” same idea that precedent remarks: refer to Fig 6, so that the reader makes the link between food process and impact on forces.
3) I find that Fig 3 is not clear. I find difficult to understand how the different elements are spatially organized. Furthermore, one does not clearly see the figure where is the distance X.
4) Line 310: the description in the text does not correspond to fig 6 (at least, I don’t understand it). The text mention primary and secondary minima and overlapping with wdW force, with does not correspond to Fig 6. Is a figure missing?
Minor remarks
5) Line 59 “ref [1]” is not in coherence with bibliography notation
6) I do not understand the reference to the “chat botté” in figure 1C (I did not see it in the text).
7) Eq (1): the terme “activity” should be defined for a2.
8) Figure 4: the indications on the figure could be clearer: the indication “attractive” in partially hindered by the curve, and the inscriptions “maximum” and “minimum” make the graph less clear. Pershaps some short arrows near the extrema could be more pleasing to the eye.
9) Eq(4): I suggest to mention the name (Poisson equation).
10) Eq(9) there is a problem in the pdf file (square instead of a letter). This formula should be checked.
11) Line 309 “… in proportion to I;is separated…” some word is missing?
Reviewer 2 Report
The review article entitled "Colloidal stability: selected examples of food colloids" proposed by La Mesa and Risuleo has an original and extravagant style, but it is not clear in the general message. From the title, a reader would expect to find an in-depth view of the literature relating to the stabilization of food colloids and the prospect of improving/modifying food stability. On the contrary, the manuscript presents some basic notions of colloidal science (surface phenomena, steric and electrostatic stabilization) in a way that can be proposed to university students and very little on food systems.
Furthermore, on 46 references cited (which is a very small number for a review), only a third have to do with food. However, the references are little discussed.
Therefore, due to the weakness of the content and clarity, I think the manuscript should be rejected.
Other points:
- Three to ten relevant keywords are needed. The authors provided over 10 keywords;
- Figure 1 makes no sense;
- Figure 7 is badly discussed;
- References are not cited according to newspaper guidelines.
Reviewer 3 Report
What follows is my reviewer report for the manuscript "colloids-872516", by Mesa and Risuleo, so far titled "Colloid Stability: Selected Examples from Food Colloids".
According to the authors, the manuscript aimed at being a review paper, in which theoretical aspects of colloidal stability are presented and, then, examples from food products are discussed. Considering this proposal, in my opinion, the document presents numerous weaknesses and drawbacks, and need a serious revision in order to become publishable. Hereafter, I present some of my main personal concerns, including both conceptual and textual ones.
1. The title of the manuscript itself may be ameliorated. The word "colloid/colloids" appears twice and, more important, the manuscript does not bring "examples from food colloids" linking the presented theory to examples, with a solid scientific argumentation. It only describes the protocol of fabrication of some cheeses. In a few words: despite the repetition of a crucial word, the title promises something that is not found within the manuscript.
2. The manuscript contains many equations, symbols ans abbreviations. As such, a list of symbols and abbreviations is of utmost importance at its beginning.
3. The section "1. Introduction" is the only one, apart the "2. Conclusions". More sections and subsections must be created in an organized way, in order to ease the readability of the text and to bring pleasantness to readers.
4. Table I seems incorrect. I think that "gas" is to be replaced by "solid" and "liquid" in the 2nd and 3rd columns, respectively. Moreover, continous and dispersed phases should be identified in this Table.
5. The content of lines 46-59, lines 115-117 and Figures 1C and 1D are, in my opinion, merely ramblings aiming to catch the attention of readers and to put it away from the superficiality of the colloids' formalism presented.
6. Table II brings only examples of cheeses. I thought that the manuscript covered a wide variety of examples of food colloids. Furthermore, in the case of cheeses, not only the fat content is relevant in the context of this manuscript, but also the contents of proteins (caseins and globular proteins) and strategic ions in such systems (e.g.: Na+, Ca2+, citrates, phosphates... and so on).
7. Since the manuscript proposes to show theoretical bases of colloid stabilisation, it must be done with scientific and theoretical rigor:
- In 1.2., equation (1) needs to be carefully derived from thermodynamics fundamentals (considering the variation dG of surface free energy when dn molecules cover a surface dA, and considering the definition of free energy...).
- In 1.3., deeper explanations and mathematical formalism associated to Eatt, [sigma]polym, and d[theta] -- only cited in eq. (2) -- need to be detailed.
- In 1.4., even though a mathematical description for the electrical double layer of colloidal particles is much better presented than in the two precedent sections, some points remain obscur: the obtaining of eq. (10) from eqs. (3) and (6) must be explained in a "softer" and detailed way.
- The quality of Figures 2, 3, and 4 is not good.
8. About sections 1.5 and 1.6:
- They are both hard to be read, because they are, in my opinion, repetitive and conceptually "shallow".
- They describe in a simplistic form the consequences of theory presented in 1.2-1.4, and present the protocol for fabrication of some cheeses. According to the title, I expected something different: several examples (covering sauces and foodstuff derived from fruits, milks, meats, and grains) in which technological aspects would be explained based on the theoretical background presented in the precedent sections.
- Once again, Figures 5, 6, and 7 have no good quality (by the way, they may have been presented in a single Figure with parts A, B, and C).
9. Section "2. Conclusions" is again quite repetitive and seems more an "abstract" than "conclusions".
- In such section, readers expect to find, clearly, answers to questions such as: what were the concrete contributions of the article to my learning on food colloids? How this review paper enhanced my understanding of theoretical aspects and state of art of a variety of food formulations with colloidal structure and properties? What would be some concrete points that are being now studied by different research teams worldwide, and how these studies are expected to clarify points not clarified in this review?
- Such conclusion could not be built, very likely due to the lack of a solid content in the body of the manuscript to generate it.
∴ Based on all of this, I think that the manuscript must be seriously reworked before a new presentation to peer-review.